# An In-Silico Comparative Study of Lipases from the Antarctic Psychrophilic Ciliate *Euplotes focardii* and the Mesophilic Congeneric Species *Euplotes crassus*: Insight into Molecular Cold-Adaptation

**DOI:** 10.3390/md19020067

**Published:** 2021-01-27

**Authors:** Guang Yang, Matteo Mozzicafreddo, Patrizia Ballarini, Sandra Pucciarelli, Cristina Miceli

**Affiliations:** School of Biosciences and Veterinary Medicine, University of Camerino, 62032 Camerino (MC), Italy; guang.yang@unicam.it (G.Y.); matteo.mozzicafreddo@unicam.it (M.M.); patrizia.ballarini@unicam.it (P.B.); cristina.miceli@unicam.it (C.M.)

**Keywords:** hydrolytic enzymes, cold-adaptation, amino acid composition, secondary structure, bioinformatics

## Abstract

Cold-adapted enzymes produced by psychrophilic organisms have elevated catalytic activities at low temperatures compared to their mesophilic counterparts. This is largely due to amino acids changes in the protein sequence that often confer increased molecular flexibility in the cold. Comparison of structural changes between psychrophilic and mesophilic enzymes often reveal molecular cold adaptation. In the present study, we performed an in-silico comparative analysis of 104 hydrolytic enzymes belonging to the family of lipases from two evolutionary close marine ciliate species: The Antarctic psychrophilic *Euplotes focardii* and the mesophilic *Euplotes crassus*. By applying bioinformatics approaches, we compared amino acid composition and predicted secondary and tertiary structures of these lipases to extract relevant information relative to cold adaptation. Our results not only confirm the importance of several previous recognized amino acid substitutions for cold adaptation, as the preference for small amino acid, but also identify some new factors correlated with the secondary structure possibly responsible for enhanced enzyme activity at low temperatures. This study emphasizes the subtle sequence and structural modifications that may help to transform mesophilic into psychrophilic enzymes for industrial applications by protein engineering.

## 1. Introduction

One of the most important factors that limits the distribution and abundance of life on Earth is the temperature. Excessively high temperatures break covalent bonds and ionic interactions between molecules, denature proteins, and destroy cell structures, with terrible consequence for living organisms adapted to live in temperate environment [1]. Conversely, low temperatures reduce biochemical reaction rates, inactive enzymes, and induce the formation of ice crystals that damages cell structures [2]. During the past decades, the extensive discovery of life at extreme thermal environments has converted our knowledge about life limitation. Perceptibly, the compatibility of those organisms with the habitat temperature is ultimately determined by their underlying genetic architecture. They must be suitably thermal adapted with the local environment as well as all their cell components [3].

The largest proportion of the biomass on earth is generated in the cold (≤5 °C). This is mainly due to the great number of microorganisms in the oceans, and other cold biomes such as the high alpine soils, terrestrial glaciers, perennially ice-covered lakes, and polar sea ice and ice sheets, in addition to the seasonally cold habitats [4]. Thereof, cold adaptation in the microbial world should be expected. To some extent, cell-specific adaptation strategies of cold adapted organisms have been identified. For example, it is known that to maintain cell membrane fluidity, psychrophiles increase the number of saturated bonds on fatty acid to introduce steric constraints that change the packing of the lipid bilayers [5]. In addition, microbes sometimes secrete ice-nucleating proteins and antifreeze-like proteins which impair ice crystal formation in the cells [6]. Most importantly, psychrophiles synthesize enzymes that efficiently work at low temperature [7].

In the last decades, research on enzymes produced by psychrophiles has exploded, as they constitute a tremendous potential in industrial application [7,8]. Psychrophilic enzymes are often characterized by high activities and reaction rates at low temperatures, and by decreased temperature stability compared to their mesophilic and thermophilic counterparts [9]. Efficient catalytic rate of psychrophilic enzyme is achieved largely by changing amino acids distribution and composition to confer increased molecular flexibility [9].

To date, many genomes from psychrophilic prokaryotes have been sequenced and exciting outcomes have been reported for various bacteria and Archaea species [10,11,12,13]. With the aid of these sequence data, it is possible to make a global identification of molecular cold adaptation [10]. However, psychrophilic eukaryotic microorganisms, including vast protozoan organisms, have been greatly ignored during this analysis.

*Euplotes spp.* are ciliated protozoa inhabiting aquatic environments, especially marine and lacustrine waters. *Euplotes focardii* is a cold-adapted species isolated from Antarctic marine sand sediments [14]. This ciliate grows optimally at temperatures close to 4 °C but does not grow at temperatures over 10 °C [15,16]. Previous studies have proved the value of *E. focardii* as model organism for cold-adaptation [17,18,19,20,21,22,23]. Recently, the entire genomes from *E. focardii* and its relative mesophilic *E. crassus* have been sequenced [24].

In a previous study [20], we reported a complete sequence comparison of a pair of *E. focardii* and *E. crassus* patatin-like lipases in order to identify residues for site-directed mutagenesis to transform the psychrophilc enzyme into the mesophilic counterpart. In the present study, we performed a comparative study of 104 hydrolytic enzymes belonging to three different lipases families from *E. focardii* and *E. crassus.* By applying bioinformatics approaches, we compared amino acid composition related to the secondary and tertiary structures to extract relevant information relative to cold adaptation. Our results not only confirm the importance of several previous recognized amino acid substitutions for cold adaptation [9], but also identify some new factors in the secondary structure possibly responsible for enhanced enzyme activity in the cold environment.

## 2. Results

### 2.1. Lipase Sequence Characterization and Analysis

By the analysis of the complete genome sequences, we identified 46 lipases from *E. focardii* and 58 lipases from *E. crassus*, which became the basic data for this investigation. Of the 46 lipases from *E. focardii*, 9 were determined to be patatin-like phospholipases, 29 αβ-hydrolase associated lipases, and 8 esterase lipases. Of the 58 lipases from *E. crassus*, 17 were identified as patatin-like phospholipases, 28 αβ-hydrolase associated lipases, and 13 esterase lipases (summarized in Appendix A). The sequence alignments revealed a degree of similarity in the range of 53–73% between the two *Euplotes* species. High similarity is relevant at the level of the conserved motives reported in Appendix A. Also the amino acid composition of the three lipases ORFs appeared very similar (Appendix A).

### 2.2. Amino Acid Composition Preferences

To evaluate the detectable trends in the amino acid composition, *E. focardii* and *E. crassus* lipases were aligned and compared. The final alignments comprised of 37,556 multiple aligned amino acid sites. Despite the high level of conservation of the amino acid frequencies, there were some differences in composition that may be symptomatic for cold adaptation (Figure 1): The strongest increasing of amino acid frequency in *E. focardii* was observed for Ser (1.43%) and for Ala (1.32%) residues. In contrast, the highest decreasing of amino acid frequencies in *E. focardii* resulted for Glu (1.07%) and Leu (2.04%).

From this dataset, the frequency of individual amino acids and property groups were also computed (Table 1). Despite the frequency of individual amino acid is fairly similar in lipases from both species (Figure 1), as indicated by *p*-values from Table 2, there were amino acid residues such as Ala, Asp, and Ser, significantly preferred in *E. focardii* with respect to *E. crassus*. On the other hand, residues Pro, Glu and Leu were significantly less favored in *E. focardii*. When comparing frequencies of occurrences of amino acid property groups, we observed that tiny and small amino acid groups were significantly preferred in *E. focardii*, whereas Glu residues were significantly avoided as shown by their corresponding *p*-values in Table 1.

### 2.3. Secondary Structural Elements

The amino acid composition of lipases of *E. focardii* and *E. crassus* based on the predicted secondary structural elements (see Section 4 Materials and Methods) are summarized in Table 2. Collectively taken, the total number of residues utilized by α-helices, β-sheets or random coils was similar in both species (*p*-value > 0.05, data not shown). However, the amino acids Glu and Leu show significantly low frequencies in the α-helices of *E. focardii* lipases (Table 2A). Furthermore, in the coil region of *E. focardii* lipases we observed that Ala, Asp, Gly, and Ser frequency is significantly high whereas Pro is significantly low (Table 2B). Except for an increase in frequency of the amino acid Ile, the *E. focardii* lipases β-sheets did not show any significant changes as compared to *E. crassus* (Table 2C).

Considering the biochemical properties of residues, there were less aliphatic and charged amino acids in the α-helices of the psychrophilic *Euplotes* (Table 2A). Except a preference of small amino acids in the coil region of *E. focardii* lipases, there were no other significant changes (Table 2B). The β-sheet regions of *E. focardii* lipases did not show any significant change compared to those from *E. crassus* (Table 2C).

### 2.4. Specific Amino Acid Substitutions

To better understand the individual contributions of the amino acid changes, we calculated the log odd scores (*LOS*) using the equations described in Section 4 Materials and Methods. Table 3 reports the *LOS_E. focardii_* values computed using Equation (1) (the *LOS_E. crassus_* scores calculated using Equation (2) showed similar results therefore are not reported in the table). The individual positive or negative values in Table 3 show that the magnitude of certain substitutions is favored or avoided, respectively. For example, the substitution of *E. crassus* Ala residues into Tyr in *E. focardii* is extremely avoided, being the *LOS* score of −11.30. In contrast, the substitution of *E. crassus* Lys into Ser in *E. focardii* is highly favored, being the *LOS* score of 9.81. In conclusion, values in Table 3 indicates that substitutions that increase the amount of Glu, Phe, Lys, and Tyr are avoided in *E. focardii* lipases with respect to those from *E. crassus*, whereas Ala, Asp, Gly, Ser, and Thr are favored.

We also analyzed amino acid substitutions in the light of the three-dimensional structures of the three lipases. To simplify this analysis, we compared a single representative member from each *E. focardii* and *E. crassus* lipase family, obtained as described under Section 4 Materials and Methods. Figure 2, Figure 3 and Figure 4 report the superimposition of *E. focardii* (light blue) and *E. crassus* (green) patatin-like phospholipases, αβ-hydrolase, and esterases, respectively. These superimpositions do not reveal significant structural differences in term of RMSD of the protein backbones including the active sites (residues in yellow in Figure 2, Figure 3 and Figure 4, unboxed panels). However, specific amino acid substitutions can be responsible for different interactions inside or between adjacent β-sheets that may interfere with the conformation of these enzymes (Figure 2, Figure 3 and Figure 4, boxed panels), evidenced in violet in the 3D-structure. In general, we found a reduction in the number and/or strength of weak bonds in the *E. focardii* lipases (Appendix A, in bold) in particular for ionic and van der Waals (VdW) interactions.

Modifications in both patatin-like phospholipases (Figure 2) such as *E. focardii* Lys^130^, Lys^177^and Thr^163^, and *E. crassus* Asp^65^ and Glu^150^ may increase the number of salt bridges or ionic interactions. However, only in *E. crassus* Tyr^260^ and Phe^31^ residues may give origin to an additional π-π stacking interactions through their aromatic side chains increasing the rigidity of the enzyme.

The *E. crassus* αβ-hydrolase shows the aminoacidic substitutions Gly^361^/Gln^368^ that can produce additional H-bonds and VdW interactions stabilizing the β-sheet (Figure 3).

Finally, *E. focardii* esterases shows Gly^72^/Thr^72^ and Asp^137^/Arg^137^ substitutions that can produce additional VdW interactions and H-bond, respectively. However, these substitutions are localized mainly at the level of the loop than in the β-sheet (Figure 4), with few effects in the structural conformation of this esterase.

### 2.5. Euplotes Lipases Codon Usage

We previously reported that the *E. focardii* genome is A/T rich [23,24] and we proposed that such A/T predilection may be a consequence of cold-adaptation: An A/T-rich genome composition can facilitate DNA strand separation and access of the polymerases to their template, and hence favor DNA replication and transcription. To investigate if A/T predilection biased the codon usage in *E. focardii* with respect *E. crassus*, we examined codon composition of three representative ORF’s from each lipase family (Appendix A). This analysis revealed that the two *Euplotes* species prefer codons with low GC content, even though in *E. focardii* the tendency is much higher.

## 3. Discussion

The objective of this study is to perform an in-silico comparison of putative lipases in two *Euplotes* species in which *E. focardii* represents a psychrophilic organism and *E. crassus* is a mesophilic counterpart. The lipases from these two *Euplotes* species fall into three main families: Patatin-like phospholipase lipases, α-hydrolase associated lipases, and esterase lipases.

Taking the advantages of bioinformatics approach to create a comparative study of lipases, we systematically analyzed the composition variation and substitution preferences of amino acids in these lipase families, which may help to unravel the potential mechanism of molecular cold adaptation. Additionally, keeping in mind that lipases are of special commercial interest, this study will contribute to protein engineering of mesophilic lipases to render them psychrophilic, or vice versa. The analysis of proteins from two phylogenically close organisms that belong to the same taxonomic group reduce the number of amino acid changes due to genetic divergence that have been an obstacle in previous similar studies. The analysis was performed at different levels, through “in-silico” characterization, amino acid compositions, Student’s t-test and, finally, by substitution patterns in the orthologous lipase proteins.

Previous attempts have been done for identifying the amino acid composition or amino acid substitution patterns. Gianese et al. [25] compared homologous structures from 7 and 21 different enzymes; Sadeghi et al. [26] compared 60 thermophilic structures and sequences with their mesophilic homologs. Furthermore, structural parameters distributions between 13 pairs of psychrophilic and mesophilic proteins were also reported [27]. However, these studies are limited by relatively small number of protein sequences taken from a wide variety of organisms. Several large-scale studies have also compared thermophile organisms with different growth temperatures to achieve a closer insight on protein thermostability at high temperatures. Some of the studies have focused on comparison within closely related lineages: Two mesophilic *Corynebacterium* species with slightly different optimum temperatures for growth, and two closely-related hyperthermophilic genera [28]. These works have detected general factors of cold adaptation. However, a large-scale comparative analysis between a strictly psychrophilic microorganism with a closely related mesophilic congeneric species was missing.

In this study, differences among *E. focardii* and *E. crassus* lipases based on their percentage amino acid compositions were found. Individual residue compositions combining with the substitution pattern in the orthologous proteins of two temperature species showed that in the psychrophilic *E. focardii* lipases there was a significant preference for small amino acid as Ala, Asp, Gly, Ser and Thr and a significant avoidance of Pro, Glu, Phe, Lys, and Leu residues (Table 1 and Table 3). This residues selection is directly correlated with cold adaptation, since it is well known that small residues increase molecular flexibility that facilitate enzyme conformational change during catalytic activity at low temperatures. Starting with the conception that aliphatic amino acids are important in maintaining conformational stability and rigidity of mesophilic enzymes, we can interpret that they are highly avoided in the helix regions of *E. focardii* lipases (Table 2A). Contrary trends are observed for the aromatic amino acids Phe and Tyr, that favor the formation of aromatic-aromatic interaction, making molecules more rigid. However, increased exposure of hydrophobic residues to the solvent enhanced protein solvation, that is considered a characteristics of cold-adapted enzymes [29]. In addition, the amino acid Pro is a highly rigid residue which will increase the stability of the protein structure [30]. Moreover, Glu and Leu residues tend to favor and stabilize the formation of helical structures [31] and therefore these residues tend to decrease molecular flexibility. Finally, the charged amino acid group residues known to contribute to ion pair electrostatic interactions that maintain conformation stability in proteins surface [29] are also significantly avoided in *E. focardii* lipases coil regions (Table 2C).

The amino acid substitution pattern with *LOS* scores indicated the most biased amino acid substitutions pairs (Table 3). In terms of involvement in significant (|*LOS*| ≥ 5) substitutions pairs, Ala is the most favorable residue in *E. focardii* lipases, as Ala is ambivalent, which can be inside or outside of the molecule. Likewise, Ala lacks a gamma-carbon, which contributes to the formation of α-helix, and increases the number of residues with small steric hindrances. This analysis also revealed which substitutions are preferred in the psychrophilic lipases shown in bold in Table 3. In this case is confirmed the tendency to change rigid amino acid such as Trp, Phe, Lys, and Tyr into small ones, i.e., Ala, Asn, Ser, and Asp. From the analysis of the 3D-strucure, we found a reduction in the number and/or strength of weak bonds in the *E. focardii* lipases. This reduction of weak bonds seems to be necessary to achieve an appropriate flexibility of the whole or crucial parts of the enzyme structure [31,32]. It is interesting to note that in this specific case of *E. focardii* lipase families, there is a common strategy adopted from these enzymes compatible with the preservation of the structural characteristics and molecular flexibility. In conclusion, the results of our analysis are in agreement with those previously reported but provide more information related to secondary and tertiary structures. Our analysis provides a base for the rational design of protein mutations in enzyme engineering to be used to broaden their spectrum of activity.

## 4. Materials and Methods

### 4.1. Sequence Collection and Analysis

The lipase genes (104 genes) extrapolated from the *E. focardii* genome [24] were locally blasted [33,34] into the *E. crassus* genome in order to identify homologues. Both genomes are available at NCBI data base under the acc. Nos. MJUV00000000.1 and MECR00000000.1, respectively. These sequences were aligned using T-coffee multiple sequence alignment program. All alignments were inspected and verified manually for a minimum cut-off score of 60% identity with all other sequences. No attempt was done to remove paralogs. The corresponding amino acid sequences of the *E. focardii* were extracted in 58 final alignments.

### 4.2. Analysis of Amino Acid Composition

To estimate and compare the amino acid composition of psychrophilic and mesophilic lipases, EMBOSS Pepstats (https://www.ebi.ac.uk/Tools/seqstats/emboss_pepstats/) was used. The amino acids were divided into 12 property groups including, acidic amino acids: Asp and Glu; aliphatic: Ile, Leu, and Val; aromatic: His, Phe, Trp, and Tyr; basic: Arg, His, and Lys; charged: Arg, Asp, Glu, His and Lys; hydrophilic: Asp, Glu, Lys, Asn, Gln, and Arg; hydrophobic: Ala, Cys, Phe, Ile, Leu, Met, Val, Trp, and Tyr; neutral: Gly, Gln, His, Ser, and Thr; non-polar: Ala, Cys, Gly, Ile, Leu, Met, Phe, Pro, Val, Trp, and Tyr; polar: Arg, Asn, Asp, Glu, Gln, His, Lys, Ser, and Thr; small: Ala, Cys, Asp, Gly, Asn, Pro, Ser, Thr, and Val; and tiny: Ala, Cys, Gly, Ser, and Thr. Some of the amino acids are included in more than one property groups. The sum of frequencies of amino acids that fall in each property group were calculated for psychrophilic and mesophilic lipases and compared. The composition data were then analyzed, and a Student’s t-test was applied to confirm significant difference between the two data sets.

### 4.3. Secondary Structure Prediction

The most common secondary structures in proteins are α-helices, β-sheets, and random coils. This analysis was intended to find out the structural parameter distribution between 58 pairs of psychrophilic and mesophilic proteins to elucidate the parameters contributing to the enzyme’s specific activity at low temperature. With this specific purpose, secondary structural elements in protein sequences were predicted using PSIPRED (http://bioinf.cs.ucl.ac.uk/psipred/). PSIPRED is a highly reliable secondary structure prediction method with ~83% reported prediction accuracy. The resulting predictions were used to compute frequencies of different amino acids and property groups of residues in three major secondary structural regions, helix (H), strand (E), and coil (C). The composition data were then analyzed, and a Student’s t-test was applied to confirm significant difference between the two data sets.

### 4.4. Amino Acid Substitution Bias

All lipase sequences from *E. focardii* were searched against genome data set of *E. crassus* and vice versa, using BLASTP with 10^−3^ expectation value cutoff and considerable length coverage. The pairwise alignments obtained from BLAST results of each lipase sequence in a query *Euplotes* species that showed best hit homolog in the subject *Euplotes* species was selected. The pairwise alignments (without gapped regions) were put in a custom Perl script to calculate amino acid substitution counts between the two lipases from respective species. The substitution counts were normalized to total amino acids present in each homolog pairs from two species and finally to all the pairs. The resultant frequency of substitutions was further used to calculate two types of likelihood log odd scores (*LOS*), as in equations are adapted from [35]:(1)$$LOS_{E.focardii}=log\frac{F\left(X_{E.focardii}\to Y_{E.crassus}\right)}{F\left(X_{E.focardii}\to Y_{E.focardii}\right)}$$
(2)$$LOS_{E.crassus}=log\frac{F\left(X_{E.crassus}\to Y_{E.focardii}\right)}{F\left(X_{E.crassus}\to Y_{E.crassus}\right)}$$
where *F*(*X_E. focardii_*→*Y_E. crassus_*) represents normalized frequency of amino acid *X* in *E. focardii* substituted by an amino acid *Y* in *E. crassus*. The *LOS* values were calculated by using background substitution frequencies among the *E. focardii* and/or *E. crassus* lipases in the denominator. The *LOS*, therefore, indicated the pattern of substitutions that are predominantly due to their thermal adaptation and therefore minimize the effect of substitutions due to any speciation events in the evolution process.

### 4.5. Tertiary Structure Prediction and Codon Usage Estimation

*E. crassus* and *E. focardii* αβ-hydrolase and esterase lipase the three-dimensional structures were obtained by homology modeling using as templates the pdb structure files 1K8Q [36] and 6A0W [37] respectively. The sequence identities between the *Euplotes* lipases and the templates were 31.34% and 30.91% for *E. crassus* and of 32.88% and 29.53% for *E. focardii,* respectively. Patatin-like phospholipases structures were obtained by a threading method using the I-Tasser web server [38] since the sequence identities with the best templates were lower than 25%. All obtained structures were finally energy minimized using the steepest descent algorithm (till the maximum force < 1000.0 kJ/mol/nm) of GROMACS tools [39], analyzed (predicting non-covalent interactions inside the protein) using the RING 2.0 web server [40], and rendered using PyMOL software (The PyMOL Molecular Graphics System, version 2.4.1 Schrödinger, LLC.).

Codon frequency (per thousand) has been estimated from three representative sequences from the three lipase families of each species using http://genomes.urv.es/CAIcal/

## Figures and Tables

**Figure 1 marinedrugs-19-00067-f001:**
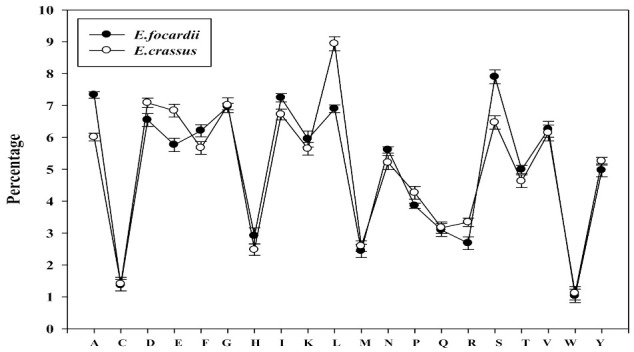
Compositional trend of individual amino acid in lipases from *E. focardii* (●) and *E. crassus* (◯).

**Figure 2 marinedrugs-19-00067-f002:**
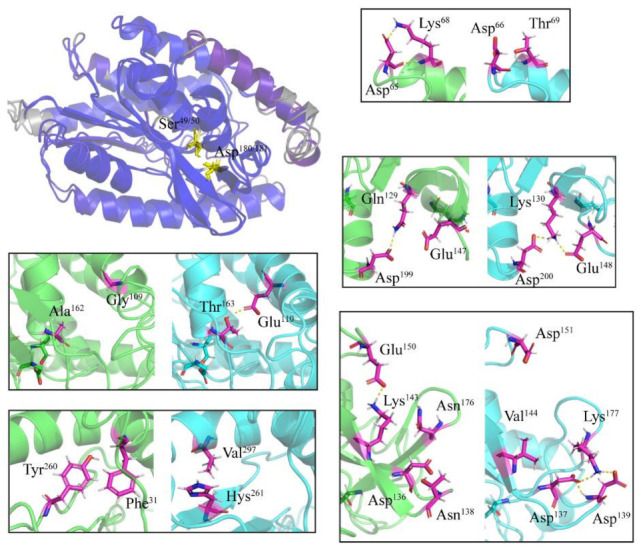
Three-dimensional structures of patatin-like phospholipases. The distances between aligned C-alpha atom pairs are colored by a color spectrum, with blue specifying the minimum pairwise RMSD and red indicating the maximum. Active site aminoacids are reported in yellow sticks. In the boxes, the amino acids differences between *E. focardii* (in light blue) and *E. crassus* (in green) are reported in violet. The models were obtained by a threading method using the I-Tasser web server.

**Figure 3 marinedrugs-19-00067-f003:**
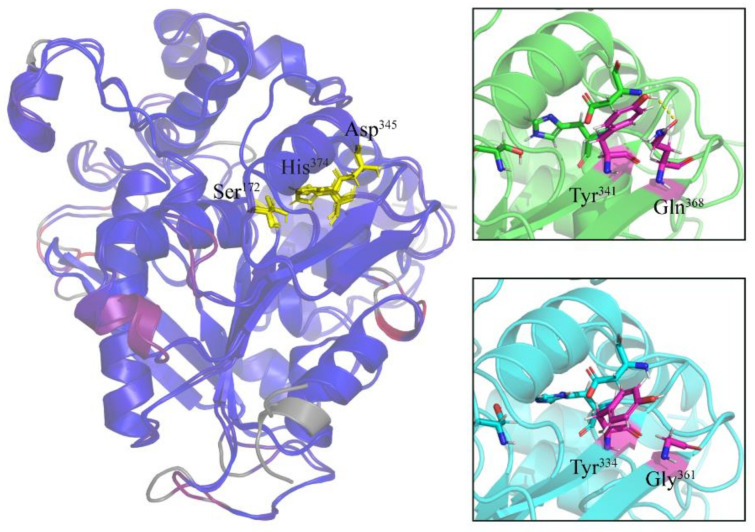
Three-dimensional structures of αβ-hydrolase. The distances between aligned C-alpha atom pairs are colored by a color spectrum, with blue specifying the minimum pairwise RMSD and red indicating the maximum. Active site amino acids are reported in yellow sticks. In the boxes, the amino acids differences between *E. focardii* (in light blue) and *E. crassus* (in green) are reported in violet. The models were obtained using as templates the PDB structure 1K8Q.

**Figure 4 marinedrugs-19-00067-f004:**
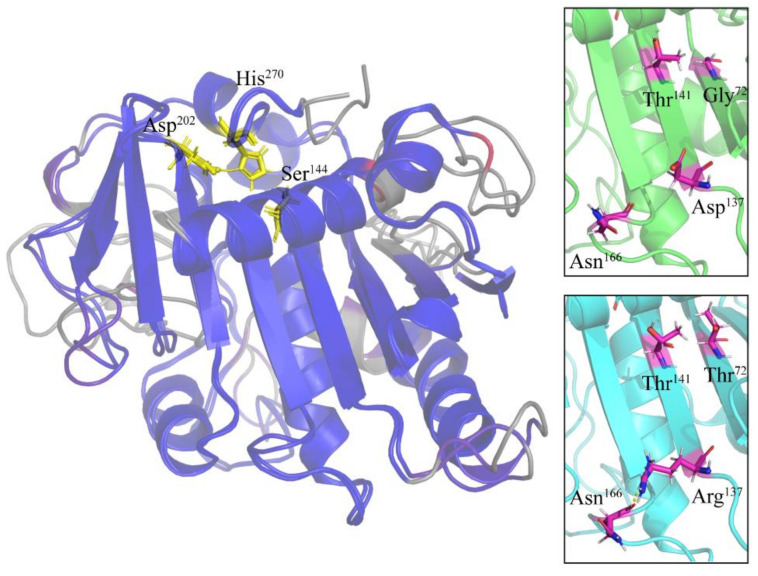
Three-dimensional structures of esterase lipases. The distances between aligned C-alpha atom pairs are colored by a color spectrum, with blue specifying the minimum pairwise RMSD and red indicating the maximum. Active site amino acids are reported in yellow sticks. In the boxes, the amino acids differences between *E. focardii* (in light blue) and *E. crassus* (in green) are reported in violet. The models were obtained using as templates the PDB structure 6A0W.

**Table 1 marinedrugs-19-00067-t001:** Frequency of individual amino acids and property groups in the lipase sequences from *Euplotes focardii* and *Euplotes crassus.* Significant differences as indicated by *t*-test *p*-values are shown in bold. ab: αβ-hydrolase associated lipases; est: Esterase lipases; pat: Patatin-like phospholipases. Avg: Average.

Amino Acids	*E. focardii*				*E. crassus*				*t*-test *p*-Value
	ab	est	pat	Avg	ab	est	pat	Avg	
A = Ala	6.7	7.2	8.1	7.3	5.1	5.6	7.3	6.0	**0.037**
C = Cys	1.0	1.9	1.1	1.4	0.9	2.1	1.2	1.4	0.713
D = Asp	6.3	6.4	7.0	6.5	7.1	6.8	7.4	7.1	**0.048**
E = Glu	5.4	5.8	6.1	5.8	6.7	6.6	7.2	6.8	**0.016**
F = Phe	6.2	6.2	6.2	6.2	5.8	6.3	4.9	5.7	0.296
G = Gly	6.3	6.6	8.1	7.0	6.7	6.7	7.6	7.0	0.875
H = His	2.7	4.5	1.6	2.9	2.4	3.6	1.4	2.5	0.199
I = Ile	7.2	7.3	7.2	7.2	6.7	6.5	6.9	6.7	0.060
K = Lys	4.8	6.4	6.6	6.0	3.6	6.8	6.6	5.6	0.583
L = Leu	8.0	5.2	7.5	6.9	10.0	7.8	9.0	8.9	**0.023**
M = Met	2.9	2.0	2.4	2.4	3.1	2.0	2.6	2.6	0.116
N = Asn	5.5	6.0	5.3	5.6	5.2	5.3	5.2	5.2	0.173
P = Pro	4.4	3.9	3.2	3.9	4.9	4.3	3.6	4.3	**0.001**
Q = Gln	3.4	3.1	2.8	3.1	3.8	3.0	2.7	3.2	0.704
R = Arg	3.4	2.3	2.4	2.7	3.8	2.6	3.7	3.3	0.170
S = Ser	7.5	8.5	7.7	7.9	6.6	6.8	6.0	6.5	**0.029**
T = Thr	5.6	5.4	4.0	5.0	5.3	4.3	4.3	4.6	0.446
V = Val	5.6	5.9	7.3	6.3	5.2	6.3	6.9	6.1	0.698
W = Trp	1.7	0.1	1.3	1.0	1.4	0.5	1.5	1.1	0.714
Y = Tyr	5.4	5.2	4.2	5.0	5.7	6.0	4.1	5.3	0.427
Amino acid property groups									
Tiny	27.1	29.6	29.0	28.6	24.6	25.5	26.5	25.5	**0.027**
Small	48.9	51.8	51.8	50.8	46.9	48.2	49.5	48.2	**0.036**
Aliphatic	20.8	18.4	21.9	20.4	22.0	20.7	22.8	21.8	0.081
Aromatic	16.0	16.1	13.2	15.1	15.3	16.4	11.9	14.5	0.346
Non-polar	55.5	51.6	56.6	54.6	55.6	54.2	55.6	55.1	0.652
Polar	44.5	48.4	43.4	45.5	44.4	45.8	44.4	44.9	0.630
Charged	22.6	25.4	23.6	23.9	23.6	26.4	26.2	25.4	0.104
Basic	10.9	13.2	10.6	11.5	9.8	13.0	11.7	11.5	0.911
Acidic	11.7	12.2	13.1	12.3	13.8	13.4	14.6	13.9	**0.022**
Hydrophilic	28.7	30.0	30.2	29.7	30.1	31.1	32.7	31.3	0.067
Hydrophobic	44.8	41.1	45.3	43.7	44.0	43.2	44.3	43.8	0.933
Neutral	25.4	28.1	24.1	25.9	24.7	24.5	22.1	23.7	0.130

**Table 2 marinedrugs-19-00067-t002:** Amino acid composition of lipases of *E. focardii* and *E. crassus* based on the predicted secondary structural elements (A. α-helix; B. coil; and C. β-sheet).

A									
Amino Acids	α-helix								*t*-test *p*-Value
	*E. focardii*				*E. crassus*				
	ab	est	pat	Avg	ab	est	pat	Avg	
A = Ala	6.9	8.5	10.3	8.6	7.0	10.0	10.0	9.0	0.498
C = Cys	1.6	0.3	1.2	1.0	1.3	0.5	1.1	1.0	0.650
D = Asp	4.1	6.8	5.5	5.5	5.1	7.5	5.3	6.0	0.304
E = Glu	5.3	6.9	7.1	6.4	5.9	7.8	7.9	7.2	**0.014**
F = Phe	8.6	7.9	5.7	7.4	7.2	8.6	3.9	6.6	0.414
G = Gly	3.0	4.1	7.7	4.9	3.7	3.7	6.7	4.7	0.731
H = His	1.7	4.5	1.2	2.5	1.7	3.3	1.2	2.1	0.442
I = Ile	9.7	7.3	8.4	8.5	9.1	7.4	8.2	8.2	0.327
K = Lys	5.1	4.5	6.6	5.4	3.9	6.0	6.4	5.4	0.994
L = Leu	13.2	12.1	10.7	12.0	14.8	13.6	12.9	13.8	**0.014**
M = Met	3.4	2.1	3.0	2.8	3.6	2.3	2.9	2.9	0.488
N = Asn	4.6	5.0	4.0	4.5	3.9	4.0	4.1	4.0	0.267
P = Pro	2.4	1.1	0.8	1.4	3.0	0.9	1.3	1.7	0.328
Q = Gln	4.4	3.9	3.6	4.0	4.7	3.6	3.2	3.9	0.565
R = Arg	3.4	2.9	4.4	3.6	3.8	2.4	4.6	3.6	0.894
S = Ser	5.4	5.7	4.7	5.3	5.4	5.2	5.4	5.3	0.851
T = Thr	5.3	6.0	3.3	4.9	4.6	2.4	3.1	3.4	0.301
V = Val	6.1	5.0	6.2	5.8	6.0	4.6	5.9	5.5	0.080
W = Trp	2.0	0.2	1.1	1.1	1.5	0.3	1.6	1.1	0.879
Y = Tyr	4.0	5.3	4.3	4.5	3.7	5.9	4.1	4.6	0.875
Amino acids property groups									
Tiny	22.1	24.6	27.2	24.6	22.1	21.7	26.4	23.4	0.279
Small	39.3	42.4	43.8	41.9	40.1	38.8	43.1	40.6	0.443
Aliphatic	28.9	24.4	25.4	26.2	29.9	25.6	27.1	27.5	**0.027**
Aromatic	16.2	17.9	12.3	15.5	14.1	18.1	10.8	14.3	0.251
Non-polar	60.7	53.7	59.5	58.0	61.0	57.8	58.6	59.2	0.524
Polar	39.3	46.3	40.4	42.0	39.0	42.2	41.4	40.9	0.527
Charged	19.6	25.7	24.9	23.4	20.3	27.0	25.5	24.3	**0.048**
Basic	10.2	12.0	12.2	11.4	9.3	11.6	12.3	11.1	0.280
Acidic	9.4	13.7	12.6	11.9	11.0	15.4	13.2	13.2	0.063
hydrophilic	27.0	30.1	31.2	29.4	27.2	31.4	31.7	30.1	0.179
hydrophobic	55.4	48.6	51.0	51.7	54.3	53.2	50.7	52.7	0.617
neutral	19.7	24.3	20.5	21.5	20.1	18.2	19.7	19.3	0.393
**B**									
**Amino Acids**	**Coil**								***t*-test** ***p*-Value**
	***E. focardii***				***E. crassus***				
	**ab**	**est**	**pat**	**Avg**	**ab**	**est**	**pat**	**Avg**	
A = Ala	5.8	5.5	6.2	5.8	4.3	4.5	5.5	4.8	**0.044**
C = Cys	0.7	2.4	0.3	1.1	0.6	3.4	1.6	1.9	0.219
D = Asp	9.8	9.1	9.8	9.6	9.2	8.0	8.7	8.6	**0.031**
E = Glu	7.1	6.0	2.4	5.2	7.3	6.8	6.2	6.8	0.280
F = Phe	4.4	3.1	5.1	4.2	4.3	3.4	5.7	4.5	0.299
G = Gly	9.4	9.6	9.8	9.6	9.1	9.1	9.5	9.2	**0.020**
H = His	2.8	4.6	1.3	2.9	2.7	3.7	2.7	3.0	0.889
I = Ile	3.9	5.1	10.2	6.4	4.3	4.3	4.5	4.3	0.381
K = Lys	3.7	7.3	1.9	4.3	3.9	7.8	6.8	6.2	0.349
L = Leu	6.1	4.9	6.2	5.7	6.1	5.8	6.0	6.0	0.578
M = Met	1.5	2.2	2.4	2.0	2.4	1.6	2.1	2.1	0.949
N = Asn	7.5	8.3	2.4	6.1	6.8	7.4	7.0	7.1	0.636
P = Pro	6.1	5.3	1.2	4.2	7.0	7.1	3.2	5.8	**0.043**
Q = Gln	2.7	3.2	0.3	2.1	3.7	2.8	2.7	3.1	0.335
R = Arg	3.4	1.7	4.4	3.2	3.6	2.2	2.8	2.9	0.706
S = Ser	9.2	9.7	10.1	9.7	8.3	8.2	8.6	8.4	**0.022**
T = Thr	5.1	3.2	7.3	5.2	5.5	4.2	4.0	4.5	0.676
V = Val	3.2	4.9	11.7	6.6	3.3	4.7	7.4	5.1	0.399
W = Trp	1.3	0.2	1.3	0.9	1.1	0.4	1.4	0.9	0.998
Y = Tyr	6.3	3.8	5.7	5.3	6.7	4.6	3.6	5.0	0.781
Amino acids property groups									
Tiny	30.2	30.4	33.6	31.4	27.8	29.4	29.2	28.8	0.120
Small	56.8	58.0	58.8	57.9	54.0	56.6	55.5	55.4	**0.047**
Aliphatic	13.2	14.9	28.1	18.7	13.6	14.8	17.8	15.4	0.435
Aromatic	14.8	11.7	13.5	13.3	14.8	12.1	13.4	13.4	0.601
Non-polar	48.7	47.0	60.1	51.9	49.1	48.9	50.4	49.5	0.571
Polar	51.3	53.0	39.9	48.1	50.9	51.1	49.6	50.5	0.569
Charged	26.9	28.6	19.9	25.1	26.7	28.5	27.2	27.5	0.447
Basic	10.0	13.5	7.6	10.4	10.2	13.7	12.3	12.0	0.380
Acidic	16.9	15.1	12.2	14.7	16.5	14.8	14.9	15.4	0.567
hydrophilic	34.2	35.5	21.2	30.3	34.5	35.0	34.2	34.6	0.433
hydrophobic	33.2	32.1	49.1	38.1	33.0	32.7	37.7	34.5	0.446
neutral	29.2	30.3	28.8	29.4	29.2	28.0	27.5	28.2	0.215
**C**									
**Amino Acids**	**β-strand**								***t*-test** ***p*-Value**
	***E. focardii***				***E. crassus***				
	**ab**	**est**	**pat**	**Avg**	**ab**	**est**	**pat**	**Avg**	
A = Ala	6.2	2.2	6.2	4.9	5.3	1.6	6.2	4.4	0.198
C = Cys	0.9	0.7	0.3	0.6	0.8	0.7	0.3	0.6	0.349
D = Asp	0.6	1.5	7.5	3.2	2.1	1.6	8.0	3.9	0.213
E = Glu	4.3	4.6	2.4	3.8	3.6	3.9	2.4	3.3	0.178
F = Phe	7.9	13.4	6.5	9.3	9.1	11.0	5.6	8.5	0.562
G = Gly	2.9	2.7	4.0	3.2	3.0	3.9	3.9	3.6	0.445
H = His	5.2	4.6	1.3	3.7	3.2	3.9	1.1	2.7	0.194
I = Ile	13.6	15.1	15.6	14.8	10.8	12.6	12.5	12.0	**0.004**
K = Lys	3.4	3.7	1.9	3.0	2.9	4.9	2.1	3.3	0.572
L = Leu	10.5	6.3	9.2	8.7	13.7	6.2	10.3	10.1	0.281
M = Met	3.7	2.7	2.4	2.9	4.8	3.0	2.9	3.5	0.139
N = Asn	1.9	1.2	2.4	1.9	2.1	0.5	2.7	1.8	0.782
P = Pro	0.1	1.5	0.3	0.6	0.1	1.3	0.3	0.6	0.592
Q = Gln	3.4	1.5	0.3	1.7	3.4	2.6	0.5	2.1	0.363
R = Arg	2.9	4.4	5.4	4.2	2.7	4.1	5.4	4.1	0.258
S = Ser	3.3	3.2	3.0	3.1	3.3	3.4	2.7	3.1	0.912
T = Thr	5.2	10.2	7.3	7.6	6.7	8.2	8.0	7.6	0.948
V = Val	15.6	11.2	17.0	14.6	13.1	14.3	17.7	15.0	0.826
W = Trp	2.9	0.2	1.3	1.5	2.3	1.1	1.5	1.7	0.764
Y = Tyr	5.2	9.2	5.7	6.7	7.1	11.0	5.9	8.0	0.136
Amino acids property groups									
Tiny	18.5	19.0	20.8	19.4	19.1	17.9	21.1	19.4	0.960
Small	36.8	34.3	48.0	39.7	36.5	35.6	49.8	40.6	0.290
Aliphatic	39.8	32.6	41.8	38.1	37.6	33.1	40.5	37.1	0.338
Aromatic	21.2	27.5	14.8	21.2	21.6	27.1	14.0	20.9	0.528
Non-polar	69.7	65.2	68.5	67.8	70.1	66.7	67.1	68.0	0.863
Polar	30.3	34.8	31.5	32.2	29.9	33.3	32.9	32.0	0.863
Charged	16.5	18.7	18.6	17.9	14.5	18.5	19.0	17.3	0.500
Basic	11.5	12.7	8.6	10.9	8.8	13.0	8.6	10.1	0.485
Acidic	5.0	6.1	10.0	7.0	5.7	5.6	10.4	7.2	0.612
hydrophilic	16.7	16.8	19.9	17.8	16.7	17.7	21.1	18.5	0.165
hydrophobic	66.7	61.1	64.2	64.0	67.0	61.5	62.8	63.8	0.748
neutral	20.0	22.1	15.9	19.3	19.6	22.1	16.2	19.3	0.803

Significant compositional differences as indicated by *t*-test *p*-values are shown in bold.

**Table 3 marinedrugs-19-00067-t003:** Log odd scores (*LOS*) of amino acid substitutions calculated using the Equation (1). Preferred and avoided residues of *E. focardii* lipases are marked with upward-pointing triangle (**Δ**) and downward-pointing triangle (**∇**), respectively. The most frequently observed replacements (|*LOS*| ≥ 5) are shown in bold.

*E. focardii*																					
		AΔ	C	DΔ	E∇	F∇	GΔ	H	I	K∇	L	M	N	P	Q	R	SΔ	TΔ	V	W	Y∇
***E. crassus***	**A**	2.43	1.28	4.21	−2.52	−0.46	0.43	1.51	−2.11	**−9.79**	−0.48	1.31	−4.21	−1.93	0.09	−1.29	1.67	0.33	−0.79	0.67	**−11.30**
	**C**	1.78	2.73	2.78	−3.61	−3.43	0.52	2.15	0.94	**−8.59**	1.09	−3.02	0.36	2.52	4.39	0.69	5.46	0.95	−0.42	3.13	**−9.40**
	**D**	3.69	−1.38	3.46	−1.55	**−6.53**	0.95	1.64	0.37	−0.51	1.38	0.94	0.91	−0.54	0.74	0.48	3.94	1.23	−0.38	0.74	**−7.34**
	**E**	**6.97**	−2.09	2.21	−2.68	−1.36	0.32	0.87	0.76	−3.72	2.85	0.63	**5.23**	0.89	−1.34	0.83	4.36	**9.55**	1.09	0.41	**−5.43**
	**F**	4.73	0.43	**7.53**	−4.22	−0.53	0.65	0.31	−0.52	**−8.96**	−0.92	0.35	−4.38	0.48	−0.46	0.76	1.29	0.32	0.25	1.94	−0.33
	**G**	2.35	−0.77	3.37	−1.03	−0.47	3.23	−4.21	0.57	−4.47	−0.59	−4.12	−0.69	−3.81	−3.31	−1.23	**5.72**	0.57	1.88	2.32	−1.48
	**H**	**5.83**	0.32	3.77	**−7.23**	**−5.65**	0.41	2.50	−1.82	**−9.46**	0.35	0.39	−3.63	−2.09	1.06	1.93	**5.83**	1.49	0.82	0.18	−2.80
	**I**	3.32	2.55	**5.93**	−4.34	**−5.95**	0.65	0.73	−3.51	−2.33	1.63	0.91	**−6.05**	1.32	−0.82	3.08	3.43	2.95	−1.28	0.84	**−6.83**
	**K**	**5.53**	1.82	1.25	**−6.53**	−4.92	2.45	−2.35	2.36	−0.38	1.37	1.02	4.38	−1.03	−0.41	**−5.61**	**9.81**	5.18	4.14	−4.22	−3.06
	**L**	4.30	2.32	3.24	−2.47	−4.81	2.62	−3.67	−3.41	−2.19	1.58	1.93	**9.43**	0.78	3.09	−4.83	2.54	3.15	4.80	−0.98	−1.36
	**M**	**5.04**	−2.08	4.21	**−5.32**	−3.19	1.31	−1.43	0.33	−0.37	1.06	2.64	−0.09	−3.96	**8.31**	0.33	3.09	0.82	0.22	5.33	−0.94
	**N**	**6.82**	3.22	**6.67**	**−6.40**	−1.62	3.51	0.44	0.75	−1.18	−3.58	−3.42	**−5.44**	1.24	−2.81	−1.27	4.36	3.56	0.32	−3.41	−2.64
	**P**	**5.38**	−3.72	4.28	−4.37	−4.71	0.58	**−5.55**	0.23	−4.78	−3.96	0.59	−2.36	0.99	**−7.03**	−0.57	3.68	**5.47**	−1.75	−1.82	−5.99
	**Q**	**8.82**	3.49	4.74	−4.51	**−5.31**	0.28	−2.39	−0.43	**−7.21**	−2.35	−1.36	**5.23**	0.39	−0.06	0.38	**6.71**	0.83	4.39	2.99	−5.69
	**R**	**6.18**	2.35	3.52	−3.62	**−6.46**	3.52	2.30	1.53	−0.57	0.33	−0.52	3.02	0.37	0.18	1.03	**6.10**	3.06	3.09	−2.30	−4.33
	**S**	**7.05**	0.64	**7.12**	−1.83	−3.67	4.12	**−5.51**	4.37	−1.16	1.22	−2.10	−2.34	−0.05	−2.26	2.08	1.05	0.97	4.59	**−8.20**	**−8.52**
	**T**	3.23	0.54	3.64	−4.64	**−7.42**	0.53	1.27	**−5.31**	−1.85	−1.29	−0.95	−0.28	3.32	0.67	2.31	2.85	3.42	−0.32	−0.58	**−6.86**
	**V**	2.06	0.73	**6.42**	**−9.45**	**−5.51**	0.48	0.31	−0.69	**−9.32**	0.31	−0.64	−2.93	0.28	−0.70	0.73	2.92	1.89	−3.22	0.72	−3.18
	**W**	**9.53**	**6.21**	3.48	**−6.47**	−3.26	4.21	0.58	3.68	**−10.60**	0.47	1.08	−4.03	0.40	3.46	−2.18	**7.53**	3.91	1.20	3.47	**−5.77**
	**Y**	3.46	0.69	2.14	−3.27	−2.53	4.74	1.91	0.84	**−6.83**	−3.04	−0.58	−1.58	−2.19	**8.95**	−1.24	**6.71**	0.39	0.33	−3.95	−1.48

## Data Availability

*Euplotes focardii* and *Euplotes crassus* genomes are available at NCBI data base under the acc. Nos. MJUV00000000.1 and MECR00000000.1, respectively.

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
