# Peer review of "An In-Silico Comparative Study of Lipases from the Antarctic Psychrophilic Ciliate *Euplotes focardii* and the Mesophilic Congeneric Species *Euplotes crassus*: Insight into Molecular Cold-Adaptation"

_marinedrugs, 2021, doi:10.3390/md19020067_

Round 1

Reviewer 1 Report

The research article emphasizes the subtle sequence and structural modifications that may help to transform mesophilic 25 into psychrophilic enzymes for industrial applications by protein. engineering.
I suggest some minor corrections before publication

  1. The discussion needs to be more elaborate and must include a more relevant article in this area.
  2. The article will be more informative if the author also includes the proposed interaction between amino acids using "3-dimensional structure analysis".
  3. I will also like to suggest to include  "codon usage and amino acid composition of ORFs" study.
    The overall article is article was well written except for some typo errors. 

Author Response

    1. The discussion needs to be more elaborate and must include a more relevant article in this area.

Response: We revised the discussion and added new references more appropriated to the cold-adaptation topic.

    1. The article will be more informative if the author also includes the proposed interaction between amino acids using "3-dimensional structure analysis".

Response: in the revised version of the paper we added an analysis of a selection of amino acid substitutions mapped over the 3D structure(from row 170).

    1. I will also like to suggest to include  "codon usage and amino acid composition of ORFs" study.

Response: We added this in the revised version of the paper (row 214)

The overall article is article was well written except for some typo errors.

Response: typos were corrected

Reviewer 2 Report

In the manuscript entitled "An in-silico comparative study of lipases from the Antarctic psychrophilic ciliate Euplotes focardii and the mesophilic congeneric species E. crassus: insight into molecular cold-adaptation.” by Yang et al. (Manuscript marinedrugs-1057252), the authors describe the bioinformatics analysis of 122 genes or proteins of both organisms in respect of their amino acid composition and the distribution of amino acids within secondary structure elements.

To the reviewer’s point of view the current version of the manuscript is well written and comprehensive. A severe point, the discussion lacks a comparison to other psychrophilic enzyme classes and the analysis that had been performed based on the primary sequence (see below).

Other more specific comments/questions:

  • In the introduction the authors write about the analysis of 122 genes or 122 hydrolytic enzymes. In the first paragraph of the results section the authors introduce 46 lipases of focardii and 58 lipases of E. crassus. If I sum up latter numbers, I end up with a total number of 104 enzymes. Please explain!
  • Line 174: I cannot follow the rationality behind the sentence. Why Phe and Tyr residues should favor the formation of salt bridges? Salt bridges can be formed between charged amino acids, most frequently Asp, Glu as well as Arg and Lys. Please comment…
  • The observations discussed in the penultimate paragraph are not appropriately discussed. Similar observations have been already made for other cold adapted proteins. For a review (page 16-17) see:

Santiago M, Ramírez-Sarmiento CA, Zamora RA, Parra LP. Discovery, Molecular Mechanisms, and Industrial Applications of Cold-Active Enzymes. Front Microbiol. 2016 Sep 9;7:1408. doi: 10.3389/fmicb.2016.01408. PMID: 27667987; PMCID: PMC5016527.

or on page 172

Collins, T., D'Amico, S., Marx, J.‐C., Feller, G. and Gerday, C. (2007). Cold‐Adapted Enzymes. In Physiology and Biochemistry of Extremophiles (eds C. Gerday and N. Glansdorff). https://doi.org/10.1128/9781555815813.ch13

Feller, G. Molecular adaptations to cold in psychrophilic enzymes. CMLS, Cell. Mol. Life Sci. 60, 648–662 (2003). https://doi.org/10.1007/s00018-003-2155-3

In light of the current literature, the authors should discuss their findings, if they are in agreement with previous findings or not. This should be discussed in the Discussion section. Moreover, I wondering, if there are also differences in the length of loop regions between E. focardii and E. crassus?

Minor points:

  • abstract, line 18: first time in use, write complete organism name in italic
  • line 31: rephrase “…life on Earth is temperature.”
  • Table 2: introduce in the caption of the table, the abbreviations “ab, est, pat, Avg”
  • Table 3A: “Helix” with small letter
  • Table 3C: correct “b-strand” to “β-strand”
  • Line 216: correct “alpha helices, beta-sheets” to “α helices and to β-sheets”

Author Response

1- In the introduction the authors write about the analysis of 122 genes or 122 hydrolytic enzymes. In the first paragraph of the results section the authors introduce 46 lipases of focardii and 58 lipases of E. crassus. If I sum up latter numbers, I end up with a total number of 104 enzymes. Please explain!

Response: the referee is right; the enzymes are only 104. We made a mistake.

2- Line 174: I cannot follow the rationality behind the sentence. Why Phe and Tyr residues should favor the formation of salt bridges? Salt bridges can be formed between charged amino acids, most frequently Asp, Glu as well as Arg and Lys. Please comment…

Response: again, the referee is right, these residues may form aromatic-aromatic interactions through their side chains. In the revised version of the manuscript, we change the statement.

3- The observations discussed in the penultimate paragraph are not appropriately discussed. Similar observations have been already made for other cold adapted proteins. For a review (page 16-17) see:

 Santiago M, Ramírez-Sarmiento CA, Zamora RA, Parra LP. Discovery, Molecular Mechanisms, and Industrial Applications of Cold-Active Enzymes. Front Microbiol. 2016 Sep 9;7:1408. doi: 10.3389/fmicb.2016.01408. PMID: 27667987; PMCID: PMC5016527.

 or on page 172

Collins, T., D'Amico, S., Marx, J.‐C., Feller, G. and Gerday, C. (2007). Cold‐Adapted Enzymes. In Physiology and Biochemistry of Extremophiles (eds C. Gerday and N. Glansdorff). https://doi.org/10.1128/

Feller, G. Molecular adaptations to cold in psychrophilic enzymes. CMLS, Cell. Mol. Life Sci. 60, 648–662 (2003). https://doi.org/10.1007/

 Responses: we discussed our results according to the suggested papers

In light of the current literature, the authors should discuss their findings, if they are in agreement with previous findings or not. This should be discussed in the Discussion section. Moreover, I wondering, if there are also differences in the length of loop regions between E. focardii and E. crassus?

Responses: according to the secondary and 3D structures, there is no significant difference in the loop regions

Minor points:

    • abstract, line 18: first time in use, write complete organism name in italic
    • line 31: rephrase “…life on Earth is temperature.”
    • Table 2: introduce in the caption of the table, the abbreviations “ab, est, pat, Avg”
    • Table 3A: “Helix” with small letter
    • Table 3C: correct “b-strand” to “β-strand”
    • Line 216: correct “alpha helices, beta-sheets” to “α helices and to β-sheets”

response: we corrected these minor points in the revised version of the paper

Reviewer 3 Report

Some issues argued by the reviewers that need to be considered in this work, and I itemized below.

  1. In order to further understand how psychrophilic enzymes efficiently work at low temperature, it’s necessary to figure out the fine-grained difference between the mesophilic and thermophilic counterparts. Not only the coarse-grained preference for changing amino acids distribution and composition, it’s especially important to identify some new features correlated with exact motifs on protein sequence or structure possibly conferring on the mechanism for high activities and reaction rates at low temperatures for industrial applications. It’s hard to clarify the factors for increasing enzyme activity and ability to cold adaptation separately. Although the proposed calculation of LOS value of amino acid substitutions seems to confine the key changes by the usage of BLAST. Most of content in result and discussion just focused on the obviously more or less favored amino acids, it’s still very hard to estimate the change for some residue to cold adaptation. I suggest the authors descript the purpose for each procedure in detail in the manuscript, and illustrate the factors directly responding to cold adaption for a complete alignment of a pair of psychrophilic and mesophilic proteins as example in the result to let reader know how to evaluate the usage for this proposed analysis for psychrophilic enzyme.
  2. It’s ambiguous for the legend of Table S2 – “there are more F and L residues were found around active site S”, “more L residues were also found in the conserved AXXXXP motif”, and “residue I is more preferable close to the active site D”. The authors did not descript how they obtain the amount of specific residues for the assigned active sites and the purpose.

Author Response

    1.  

In order to further understand how psychrophilic enzymes efficiently work at low temperature, it’s necessary to figure out the fine-grained difference between the mesophilic and thermophilic counterparts. Not only the coarse-grained preference for changing amino acids distribution and composition, it’s especially important to identify some new features correlated with exact motifs on protein sequence or structure possibly conferring on the mechanism for high activities and reaction rates at low temperatures for industrial applications. It’s hard to clarify the factors for increasing enzyme activity and ability to cold adaptation separately. Although the proposed calculation of LOS value of amino acid substitutions seems to confine the key changes by the usage of BLAST. Most of content in result and discussion just focused on the obviously more or less favored amino acids, it’s still very hard to estimate the change for some residue to cold adaptation. I suggest the authors descript the purpose for each procedure in detail in the manuscript, and illustrate the factors directly responding to cold adaption for a complete alignment of a pair of psychrophilic and mesophilic proteins as example in the result to let reader know how to evaluate the usage for this proposed analysis for psychrophilic enzyme.

Response: in a previous paper (Yang G., De Santi C., de Pascale D., Pucciarelli S., Pucciarelli S., Miceli C. Characterization of the first eukaryotic cold-adapted patatin-like phospholipase from the psychrophilic Euplotes focardii: Identification of putative determinants of thermal-adaptation by comparison with the homologous protein from the mesophilic Euplotes crassus. Biochimie. 2013 Sep;95(9):1795-806.), we reported a complete alignment of a pair of psychrophilic and mesophilic patatin like lipases in order to identify residues for site-directed mutagenesis and biochemical characterization. The goal of this paper is different: we compared amino acid composition related to secondary and tertiary structures to extract relevant information that is statistically valid relative to cold adaptation. We better explain the purpose in the revised version of the paper. (line 77)

    1.  

It’s ambiguous for the legend of Table S2 – “there are more F and L residues were found around active site S”, “more L residues were also found in the conserved AXXXXP motif”, and “residue I is more preferable close to the active site D”. The authors did not descript how they obtain the amount of specific residues for the assigned active sites and the purpose.

Response: We removed the comment of table S2 since it was related to a complete sequence alignment and analysis, therefore not related to this table. Furthermore, in the revised version of the paper, we explained that the active sites were identified by mapping the amino acid sequences to the 3D structure. Thanks to the referees’ comment, we realized that there were some mistakes in the active site sequences that are now corrected in the new version.

Round 2

Reviewer 2 Report

The revised version has imporved. I just have one more minor point. The authors discuss in more detail structures of lipases. It would be great, if the authors could include for each structure the ProteinDataBank (PDB) identifier, to clearly identify to which structure is referred to. These includes the figure captions of Figure 2, Figure 3, as well as Figure 4.

Author Response

The revised version has imporved. I just have one more minor point. The authors discuss in more detail structures of lipases. It would be great, if the authors could include for each structure the ProteinDataBank (PDB) identifier, to clearly identify to which structure is referred to. These includes the figure captions of Figure 2, Figure 3, as well as Figure 4.

Response: in the revised version of the paper, we included these information in the figure captions. Furthermore, we revised the corresponding chapter in the Material and Methods. All the modification were performed using the “track change” function.

Reviewer 3 Report

The authors have addressed the points raised in my previous review, and publication of the manuscript as is recommended.

Author Response

The authors have addressed the points raised in my previous review, and publication of the manuscript as is recommended.

Response: We thank the Reviewer for the suggestions that greatly improved our manuscript.